# Association between the risk of malnutrition and functional capacity in patients with peripheral arterial disease: A cross-sectional study

Juliana Carvalho[1], Marilia A. Correia[2], Hélcio Kanegusuku[1], Paulo Longano[2], Nelson Wolosker[1], Raphael M. Ritti-Dias[2], Gabriel Grizzo Cucato[1,3]*

1 Hospital Israelita Albert Einstein, São Paulo- SP, Brazil, 2 Universidade Nove de Julho (UNINOVE), São Paulo- SP, Brazil, 3 Northumbria University, Newcastle upon Tyne, United Kingdom

* gabriel.cucato@northumbria.ac.uk

## Abstract

**Data Availability Statement:** All data are available with the paper and Supporting information files.

### Introduction

The risk of malnutrition is an important predictor of functional capacity in the elderly population. However, whether malnutrition is associated with functional capacity in patients with peripheral artery disease (PAD) is poorly known.

### Purpose

To analyse the association between the risk of malnutrition and functional capacity in patients with PAD.

### Methods

This cross-sectional study included 135 patients with PAD of both genders, ≥50 years old, with symptomatic PAD (Rutherford stage I to III) in one or both limbs and with ankle-brachial index ≤0.90. The risk of malnutrition was assessed by the short form of the Mini Nutritional Assessment-Short Form and patients were classified as having normal nutritional status (n = 92) and at risk of malnutrition (n = 43). Functional capacity was objectively assessed using the six-minute walking test (6MWT, absolute maximal distance and relativized and expressed as a percentage of health subjects), short-physical performance battery (SPPB, balance, gait speed and the sit and stand test) and the handgrip test, and subjectively, using the Walking Impairment Questionnaire and Walking Estimated-Limitation Calculated by History. The association between the risk of malnutrition and functional capacity was analysed using bivariate and multivariate logistic regression adjustments for gender, age, ankle-brachial index, body mass index, use of statins, coronary arterial disease and stroke. For all statistical analyses, significance was accepted at p<0.05.

**Funding:** GGC holds a grant from The National Council for Scientific and Technological Development (CNPq #409707/2016-3).

**Competing interests:** The authors have declared that no competing interests exist.

## Results

Thirty-two per cent of our patients were classified with a risk of malnutrition. The risk of malnutrition was associated with the absolute 6MWT total distance (OR = 0.994, P = 0.031) relative 6MWT total distance (OR = 0.971, P = 0.038), lowest SPPB total score (OR = 0.682, P = 0.011), sit and stand (OR = 1.173, P = 0.003) and usual 4-meter walk test (OR = 1.757, P = 0.034).

## Conclusion

In patients with PAD, the risk of malnutrition was associated with objective measurements of functional capacity.

## Introduction

Peripheral artery disease (PAD) is characterised by a systemic arteriosclerotic process, which results in partial or total obstruction in the arteries of the lower limbs [1]. The most common symptom of PAD is claudication [2], consisting of pain or cramp during walking that is relieved at rest. The symptoms of claudication affect about 20 to 50% of patients with PAD, leading to reduced levels of physical activity [3], functional capacity [4] and quality of life [5].

Impaired nutritional status has been considered an additional risk factor for the severity of the PAD [6, 7]. A study by Thomas et al. [8] observed that approximately 78% of patients admitted for vascular surgery were classified as malnourished. Additionally, another study [7] found that 38% of patients submitted to endovascular surgery were malnourished. The risk of malnutrition can be evaluated through Mini Nutritional Assessment-Short Form (MNA-SF), a valid and simple nutritional screening tool [9, 10], which can be easily applied in clinical settings. The MNA-SF consists of six items related to food intake, weight loss, mobility, stress or acute illness, neuropsychological disorders and body-mass index values. The questionnaire score ranges from 0 to 14 points, and individuals are classified as: malnourished (MNA-SF score ≤7), at risk of malnutrition (MNA-SF score ≥8 ≤11) or normal nutritional status (MNA-SF score ≥ 12) [9, 11].

Interestingly, in previous studies risk of malnutrition, an intermediate classification of nutritional status, was associated with reduced functional capacity and lower limb strength in healthy elderly [12] and patients with long-term conditions such as stroke [13], renal failure [14], diabetes [15] and chronic obstructive pulmonary disease [16]. Thus, this study aimed to analyse the association between the risk of malnutrition and functional capacity in patients with symptomatic PAD. Our hypothesis is that malnutrition has an additional factor to functional impairments.

## Methods

### Study design

This observational cross-sectional study follows the Strengthening the Reporting of Observational Studies in Epidemiology (STROBE) checklist [17]. Functional capacity was assessed using objective tests (six-minute walk test, Short-Physical Performance Battery and handgrip strength) and subjective tools such as the Walking Impairment Questionnaire (WIQ) and Walking Estimated-Limitation Calculated by History (WELCH). The sample was evaluated

according to nutritional status using MAN-SF and was classified as "at risk of malnutrition" and "normal nutritional status". The functional capacity parameters were compared between groups.

## Sample and data collection

Patients were recruited at a tertiary center specializing in vascular disease in São Paulo—Brazil. Data collection was carried out between September 2015 and October 2019. All patients were instructed regarding the experimental procedures and signed informed written consent before participation. This study was approved by the ethics committee of Hospital Israelita Albert Einstein, Brazil and Hospital das Clinicas, University of Sao Paulo, Brazil.

We included patients of both genders, ≥50 years old, with symptomatic PAD (Rutherford stage I to III) in one or both limbs, ankle-brachial index ≤0.90 [18]. Patients with non-compressible vessels, amputated limbs and/or ulcers, and low cognitive levels (<17 of the Montreal Cognitive Assessment) [19] were excluded.

## Clinical data

A standardised interview was conducted, including an evaluation of sociodemographic information, such as age and gender (male or female) and conditions of comorbidities (doctor-diagnosed history and medications). Current smoking, obesity (body mass index ≥30 kg/m$^2$), diabetes (doctor-diagnosed or use of drugs), hypertension (doctor-diagnosed or antihypertensive drugs), dyslipidaemia (doctor-diagnosed or use of medication) and coronary heart disease (doctor-diagnosed or use of drugs) were assessed.

## Dependent variable: Risk of malnutrition

The risk of malnutrition was assessed through the MNA-SF [10], which consists of six questions based on conditions of self-visualization of food intake (0 to 2 points), weight loss (0 to 3 points), mobility (0 to 2 points), psychological stress (0 or 2 points), neuropsychological problems (0 to 2 points) and a measure of body mass index (0 to 3 points). The sum of the points provides scores ranging from 0 to 14. Patients were classified as: ≤ score 7 as "malnourished", score 8 to 11 as "at risk of malnutrition", and score ≥ 12 as "normal nutritional status" [15].

## Independent variables

**Objective measurements of functional capacity.** *The six-minute walk test.* The 6MWT [20] consists of walking for six minutes in a 30-meter long flat corridor, and patients were encouraged to "walk at the usual pace" and instructed to rest when necessary. The 6MWT total distance was defined as the maximum distance achieved by the patients at the end of the test. In addition, the 6MWT total distance was relativised based on the results of 6MWT performed by healthy individuals using Brito's et al. equation [21], previously used in patients with PAD [22].

$$6MWDpred = 890.46 - (6.11 \times age) + (0.0345 \times age^2) + (48.87 \times gender) - (4.87 \times BMI)$$
$$(\text{where male gender} = 1 \text{ and female gender} = 0)$$

*Short Physical Performance Battery.* The SPPB [23] comprises a group of tests involving balance, gait speed and the sit and stand test. The balance consisted of the patient remaining in each timed foot position for 10 seconds (feet side by side, semi-tandem and tandem), and the evaluator demonstrated each position. The gait speed consisted of the patient walking for 4 meters twice in a usual and fast way, being the fastest time used for the analysis. The sit and

stand test required the initially seated patient to get up from the chair five times with arms flexed over the chest as quickly as possible, and time recorded. Each test score ranged from 0 to 4, and the total score was calculated by adding scores of three tests, ranging from 0 to 12, being 0 in the worst function and 12 in the best function [24].

*Handgrip Strength Test.* The handgrip strength test was obtained through isometric contractions using a digital dynamometer (EH101, Camry, USA) adjusted and calibrated on a scale from 0 to 100 kgf. The patient was seated with feet resting on the ground, and elbows flexed to 90 degrees and forearms and wrists in a neutral position. Three maximum voluntary contractions of five seconds were performed in both arms with an interval of one minute between each attempt. We considered the highest value for the analysis [25].

**Subjective measurements of functional capacity.** *Walking Impairment Questionnaire.* The WIQ [26] is an instrument that provides self-reported indicators of the walking capacity of patients with PAD and claudication symptoms in different situations, such as walking distance, walking speed and ability to climb stairs. The total score ranges from 0 to 100, where 0 represents extreme limitation, and 100 represents no walking difficulties.

*Walking Estimated-Limitation Calculated by History.* The WELCH [27] is a questionnaire that presents four questions related to the speed and time the patient can walk compared to relatives, friends or individuals of the same age without PAD. The total score ranges from 0 to 100, with 0 indicating a patient who can walk for 30 seconds slower than relatives, friends or colleagues in the same age group, and a score of 100 indicates who can walk for three hours compared to people in the same age group.

## Statistical analysis

We describe the data in median (interquartile interval) or frequency. The association between the risk of malnutrition and functional capacity was analysed using bivariate and multivariate logistic regression analysis with adjustment for gender, age, ankle-brachial index, body mass index, use of statins, coronary arterial disease and stroke, which are classical confounders in PAD [28, 29]. The $p < 0.05$ value was considered significant. All statistical analyses were performed with SPSS version 25.0 (IBM Corporation, SPPS Inc, Chicago, IL).

## Results

Three hundred and two patients were recruited. However, 31 patients were excluded because they did not answer the MNA-SF questionnaire, and 112 patients were due to the low score on the cognitive assessment, since these patients were probably not able to answer the questionnaires correctly, and this could be a confounding fact in the analyses and 21 excluded for not performing the 6MWT. Furthermore, only three patients were classified as malnourished and were excluded due to the insufficient sample size. Thus, we analysed the data of 135 patients, 68% of patients were classified as having normal nutritional status, and 32% were classified as at risk of malnutrition. The flowchart of the study is shown in Fig 1.

The characteristics of the patients with normal nutritional status and risk of malnutrition are presented in Table 1.

The characteristics and prevalence of risk factors were similar between groups. Patients at risk of malnutrition presented more prevalence of psychological stress/acute diseases (P = 0.013), neuropsychological problems (P = 0.022) and a lower BMI classification (P = 0.005).

Table 2 shows the association between the risk of malnutrition and functional parameters in PAD patients.

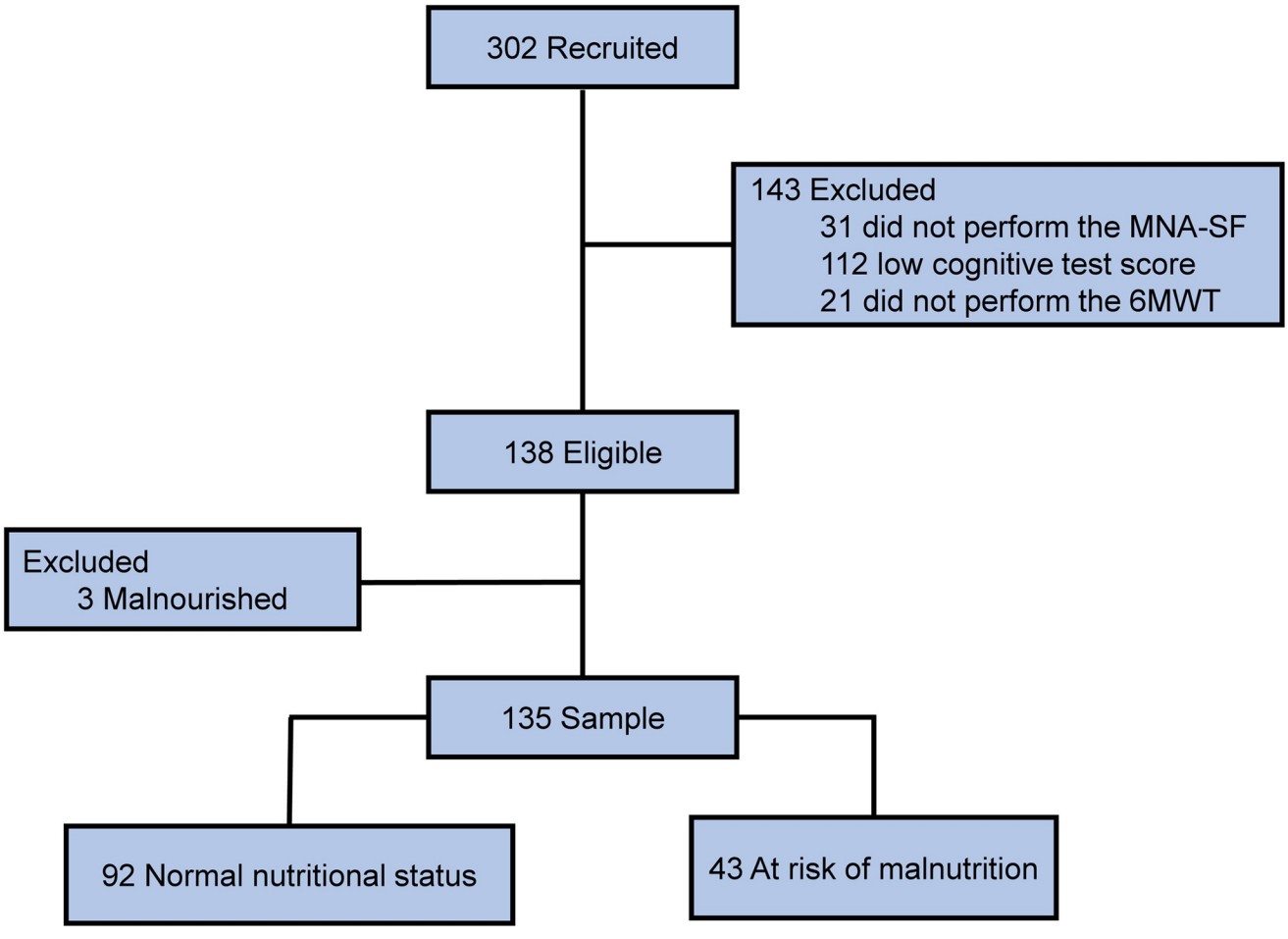

**Fig 1. Flowchart of the study.**

We observed a significant association between the risk of malnutrition and functional capacity after adjustments in absolute (OR = 0.994; P = 0.031) and relative (OR = 0.971, P = 0.038) values of 6MWT, SPPB (OR = 0.682; P = 0.011) sit and stand test (OR = 1.173, P = 0.003), usual 4-meter (OR = 1.757, P = 0.034).

## Discussion

The main findings of this study were; a) 32% of our sample were classified at risk of malnutrition, and; b) the risk of malnutrition was associated with lower walking distance and lower limb strength.

In the present study, we used the MNA-SF questionnaire to assess the risk of malnutrition in PAD patients. This questionnaire has been used in several populations, such as healthy individuals and patients with different chronic diseases [15, 16, 30]. Still, until the present study, MNA-SF was not explicitly used in PAD. Using this questionnaire, we demonstrated that 32% of our patients were classified as at risk of malnutrition. These values are similar to those observed in patients with diabetes mellitus [15] and heart failure [30], with a prevalence of 33% and 30%, respectively.

**Table 1. Clinical characteristics of patients with peripheral arterial disease associated with the risk of malnutrition n = 135.**

| Variables | N | Normal nutritional status | N | Risk of malnutrition | P |
|---|---|---|---|---|---|
| Age (years) | 92 | 65 (11) | 43 | 64 (10) | 0.780 |
| Sex (men, %) | 92 | 68 | 43 | 53 | 0.092 |
| Weight (kg) | 92 | 74 (19) | 43 | 72 (18) | 0.269 |
| Body mass index (kg/m$^2$) | 92 | 27 (5) | 43 | 27 (9) | 0.654 |
| Ankle/brachial index | 92 | 0.60 (0.24) | 43 | 0.59 (0.27) | 0.850 |
| **Risk Factors (%)** | | | | | |
| Smoke | 91 | 20 | 43 | 22 | 0.763 |
| Hypertension | 92 | 83 | 43 | 81 | 0.769 |
| Diabetes Mellitus | 92 | 54 | 42 | 53 | 0.893 |
| Dyslipidemia | 92 | 76 | 43 | 81 | 0.426 |
| Coronary disease | 89 | 36 | 41 | 32 | 0.580 |
| Stroke | 89 | 9 | 41 | 16 | 0.204 |
| Cancer | 86 | 12 | 43 | 13 | 0.849 |
| Revascularization | 86 | 19 | 41 | 12 | 0.292 |
| Heart failure | 88 | 12 | 39 | 12 | 0.960 |
| **Medications (%)** | | | | | |
| Statins | 73 | 91 | 32 | 81 | 0.118 |
| Vasodilators | 73 | 34 | 32 | 31 | 0.764 |
| Antiplatelet | 73 | 86 | 32 | 81 | 0.507 |
| Beta blockers | 73 | 50 | 32 | 37 | 0.213 |
| Diuretics | 73 | 39 | 32 | 50 | 0.266 |
| ACE inhibitors | 73 | 24 | 32 | 18 | 0.504 |
| ARA | 73 | 29 | 32 | 25 | 0.691 |
| **MNA- SF components** | | | | | |
| Food Intake (0,1,2) | 92 | 2 (0) | 43 | 2 (1) | 0.000 |
| Weight loss (0,1,2,3) | 92 | 3 (0) | 43 | 1 (2) | 0.000 |
| Mobility (0,1,2,3) | 92 | 2 (0) | 43 | 2 (0) | 0.196 |
| Psychological/ acute disease (0,2) | 92 | 2 (0) | 43 | 2 (2) | **0.013**[*] |
| Neurophysiological (0,1,2) | 92 | 2 (0) | 43 | 2 (0) | **0.022**[*] |
| BMI classification (0,1,2,3) | 92 | 3 (0) | 43 | 3 (1) | **0.005**[*] |

The values are presented as median (interquartile range) or relative frequency. BMI—body mass index.; ACE = angiotensin-converting-enzyme; ARA = angiotensin receptor antagonist.

We demonstrated that the risk of malnutrition was associated with objective measurement of functional capacity analysed by the absolute and relative six-minute walking test, usual 4-meter and sit and stand test, independently of classical PAD confounders. These results demonstrated that nutritional status is related to walking distance and lower limb strength, both crucial components of overall health in PAD patients [31]. Our study did not examine the possible physiological mechanisms, but some hypotheses can explain these associations. Evidence indicates that inadequate nutrition may favour the progression of inflammation in the epithelium [32], due to high blood concentrations of LDL [33] and changes in the immune system [34], such as the release of cytokines and chemokines [35] that contribute to accelerating the atherosclerotic narrowing of the arteries. In addition, low intake of nutrients, especially vitamin D [36], fibers and antioxidants can promote mitochondrial dysfunction, leading to an alteration in ATP synthesis [37], causing impairment in muscle oxygen perfusion [38], altering

**Table 2. Logistic regression bivariate and multivariate modelling, associations between at risk of malnutrition and functional parameters in PAD participants.**

| Independent Variables | Bivariate Model | | | | Adjusted Model | | | |
|---|---|---|---|---|---|---|---|---|
| | N | OR | CI 95% | P | N | OR | CI 95% | P |
| Absolute 6MWT total distance, m | 135 | 0.996 | 0.993; 1.000 | 0.056 | 101 | 0.994 | 0.989; 0.999 | **0.031**\* |
| Relative 6MWT total distance, % | 135 | 0.983 | 0.963: 1.003 | 0.103 | 101 | 0.971 | 0.944: 0.998 | **0.038**\* |
| SPPB, total score | 124 | 0.783 | 0.638; 0.961 | **0.019**\* | 91 | 0.682 | 0.509; 0.915 | **0.011**\* |
| Sit and stand 5 times, sec | 124 | 1.098 | 1.031; 1.170 | **0.003**\* | 91 | 1.173 | 1.056; 1.304 | **0.003**\* |
| 4-meter usual walk, m/s | 124 | 1.410 | 1.016; 1.958 | **0.040**\* | 91 | 1.757 | 1.043; 2.959 | **0.034**\* |
| 4-meter fast walk, m/s | 124 | 1.336 | 0.903; 1.977 | 0.148 | 91 | 1.535 | 0.888; 2.652 | 0.125 |
| Handgrip strength, kgf | 133 | 0.980 | 0.947; 1.014 | 0.248 | 100 | 0.978 | 0.936: 1.022 | 0.330 |
| WIQ distance, score | 133 | 0.993 | 0.977; 1.010 | 0.434 | 101 | 0.987 | 0.966; 1.009 | 0.260 |
| WIQ speed, score | 133 | 0.983 | 0.958; 1.009 | 0.204 | 101 | 0.983 | 0.949; 1.017 | 0.322 |
| WIQ stars, score | 133 | 0.992 | 0.978; 1.007 | 0.992 | 101 | 0.994 | 0.975; 1.014 | 0.578 |
| Total WIQ, score | 133 | 0.990 | 0.964; 1.008 | 0.213 | 101 | 0.982 | 0.953; 1.013 | 0.259 |
| WELCH total, score | 135 | 0.983 | 0.972; 1.008 | 0.265 | 101 | 0.987 | 0.963; 1.011 | 0.274 |

SPPB- Short Physical Performance Battery, WIQ -Walking Impairment Questionnaire, WELCH—Walking Estimated-Limitation Calculated by History. Adjusted model. For gender: Age, ankle-brachial index, body mass index, statins use, diabetes mellitus, coronary arterial disease and stroke.

skeletal muscle function in density, contractility and strength in the lower limbs, which would contribute to the greater functional decline [39].

In the present study, we did not observe the association between the risk of malnutrition with subjective measures of functional capacity using a specific questionnaire for PAD patients such as WIQ and WELCH. One possible explanation is that the subjective method may underestimate the values of functional capacity when compared to objective methods [40]. Furthermore, physical exertion performed in objective methods of function capacity (such as 6MWT and gait speed) can differ from the patient's daily activity. This might explain the lack of association with self-perception of PAD-induced walking impairments.

Regarding practical implications, our results may draw attention to healthcare providers to determine the nutritional status of patients with PAD, since we observed a high prevalence of risk of malnutrition and being at risk of malnutrition can lead to a significant decline in walking capacity and lower limb strength. As a result, the MNA-SF could be easily applied in clinical practice to identify patients at risk of malnutrition with time efficiency (with an average application time of three minutes), helping to decide on better treatment strategies (nutrition, exercise, etc.) for these patients.

This study has some limitations. This is a cross-section study that does not allow us to establish causality. Due to the small number of cases, malnutrition was not analysed, which could provide information on the magnitude of the outcomes. The use of self-reported assessments is susceptible to information bias.

In conclusion, the risk of malnutrition was associated with lower functional capacity and lower limb strength. These results suggest that assessment of nutritional status could help define therapeutic approaches in symptomatic PAD patients.

## Supporting information

**S1 Data.**
(XLSX)

## Author Contributions

**Conceptualization:** Juliana Carvalho, Nelson Wolosker, Raphael M. Ritti-Dias, Gabriel Grizzo Cucato.

**Data curation:** Juliana Carvalho, Marilia A. Correia.

**Formal analysis:** Marilia A. Correia, Raphael M. Ritti-Dias, Gabriel Grizzo Cucato.

**Funding acquisition:** Gabriel Grizzo Cucato.

**Investigation:** Juliana Carvalho, Marilia A. Correia, Hélcio Kanegusuku, Paulo Longano.

**Project administration:** Raphael M. Ritti-Dias, Gabriel Grizzo Cucato.

**Software:** Paulo Longano.

**Supervision:** Raphael M. Ritti-Dias, Gabriel Grizzo Cucato.

**Validation:** Marilia A. Correia, Hélcio Kanegusuku, Paulo Longano, Nelson Wolosker, Raphael M. Ritti-Dias, Gabriel Grizzo Cucato.

**Visualization:** Marilia A. Correia, Hélcio Kanegusuku, Paulo Longano, Nelson Wolosker, Gabriel Grizzo Cucato.

**Writing – original draft:** Juliana Carvalho.

**Writing – review & editing:** Marilia A. Correia, Hélcio Kanegusuku, Paulo Longano, Nelson Wolosker, Raphael M. Ritti-Dias, Gabriel Grizzo Cucato.

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
