## [Decision Letter · Decision Letter 0]

7 Apr 2022

PONE-D-22-05299Association between the risk of malnutrition and functional capacity in patients with peripheral arterial disease: A cross-sectional studyPLOS ONE

Dear Dr. Cucato,

Thank you for submitting your manuscript to PLOS ONE. After careful consideration, we feel that it has merit but does not fully meet PLOS ONE’s publication criteria as it currently stands. Therefore, we invite you to submit a revised version of the manuscript that addresses the points raised during the review process. Please have a professional editing service to edit your manuscript before the resubmission of your manuscript.

We look forward to receiving your revised manuscript.

Kind regards,

Yih-Kuen Jan, PhD

Academic Editor

PLOS ONE

Journal Requirements:

4. Check the last line of the abstract to ensure it is the same.  The reason for this check is to ensure that the AEs and Reviewers are sent correct information to allow them to make a good decision on whether they can manage/review the manuscript.  Only send back for a change if the abstract on EM and in the manuscript are VASTLY different.

Reviewers' comments:

Reviewer's Responses to Questions

**Comments to the Author**

1. Is the manuscript technically sound, and do the data support the conclusions?

Reviewer #1: Yes

Reviewer #2: Yes

2. Has the statistical analysis been performed appropriately and rigorously? 

Reviewer #1: Yes

Reviewer #2: Yes

3. Have the authors made all data underlying the findings in their manuscript fully available?

Reviewer #1: Yes

Reviewer #2: Yes

4. Is the manuscript presented in an intelligible fashion and written in standard English?

Reviewer #1: Yes

Reviewer #2: Yes

5. Review Comments to the Author

Reviewer #1: 1. The authors are commended for their work.

2. I still continue to see the term "intermittent claudication" used in some (but not all) vascular publications. The nomenclature using the term "intermittent" seems at least to this reviewer, to be outdated. The ACC/AHA guidelines do not refer to it in this way and we don't typically refer to angina as "intermittent angina" so consider just referring to the symptom as "claudication."

3. Abstract, line 40: Revise the sentence "We considered as significant p<0.05" as it is awkward as written.

4. Abstract, line 42: Do the authors mean distance achieved during the six minute walk test, claudication onset time? Please clarify absolute and relative as well, typically six minute walk test distance is a key performance outcome often reported.

5. Abstract, line 43: Lower SPPB total score?

6. Risk of malnutrition and peripheral arterial disease are already in the title so consider different keywords.

7. Introduction, line 62: Minor but change "lower limbs strengths" to "lower limb strength".

8. Were other comorbidities exclusionary? For example, patients with heart failure, previous cardiac and/or peripheral revascularization, those taking medications to treat PAD (cilostazol, etc) were excluded? These and other comorbidities could have had an impact on functional testing so please clarify or perform analyses controlling for these variables.

9. For the six minute walk test, why were patients asked to complete it at their usual pace, rather than as fast as they could?

10. Discussion, lines 225-226: Please revise "worst perfusion".

11. Table 1: Minor, but there is a typo (Phycological).

12. I appreciate Table 2 and the information it provides, but I think it would be helpful for readership to provide a figure(s) to graphically display some of the data if possible.

13. The paper would benefit from additional copy editing for the English language.

Reviewer #2: The manuscript studies the association between the risk of malnutrition and functional capacity in patients with peripheral arterial disease and claudication. The disease process and associated co-morbidities can lead to poor quality diet and low levels of functional capacity sometimes complete immobility, with resultant energy, protein, and micronutrient deficiencies. This research attempts to emphasize that a proper diet maybe improve the functional capacity of patients with peripheral arterial disease and claudication, which implies dietetic care plays a vital role in the management of PAD. Before I can recommend it for publication, the following questions and comments should be addressed.

1)An review published in 2020 year demonstrated that most patients with PAD are overweight or obese, 3/4 under sub-optimal nutritional status and high-fat mass, lower vitamins, and minerals. I suggest the authors explain the definition of malnutrition in people with PAD clearly.

2)I notice that the SPPB is consist of a series test. I suggest the authors explain it in the abstract section briefly. Now, the methods part and the results part are not a one-to-one correspondence.

3)The authors only concluded the association between the risk of malnutrition and objective measurements of functional capacity in patients with peripheral arterial disease and claudication. How about the association between the risk of malnutrition and subjective measurements of functional capacity in patients with peripheral arterial disease and claudication?

4)For the relationship between functional capacity and PAD, I suggest the authors give an explicit explanation.

5)For the relationship between nutritional status and PAD, I suggest the authors give an explicit explanation.

6)For the applicability of MNA-SF in brazil’s PAD, I suggest the authors give an explicit explanation.

7)In lines 72-79, it is not clear why the authors want to invest the association between the risk of malnutrition and functional capacity in patients with peripheral arterial disease and claudication. I suggest the authors give an explicit explanation.

8)In lines 99-101, the inclusion criteria are inconsistent with the abstract section. How about the assessment of claudication？

9)In lines 202-206, the results section, it is not clear the association between the risk of malnutrition and subjective measurements of functional capacity (WIQ, and WELCH) in patients with peripheral arterial disease and claudication, and the balance with the risk of malnutrition.

10)In lines 225-228, it is not clear how nutritional deficiency aggravates the functional capacity in people with PAD and claudication.

11)In lines 244-246, there are some wrong descriptions. it is not clear whether functional capacity is due to nutritional deficiency in this study.

12)In addition, I suggest the authors improve the presentation by considering the following minor changes:

Line 40, “per cent” -> “percent”,

Line 73, the questionnaire used in references 13 and 14 should be MNA instead of MAN-SF.

6. PLOS authors have the option to publish the peer review history of their article (what does this mean?). If published, this will include your full peer review and any attached files.

Reviewer #1: No

Reviewer #2: **Yes: **Yameng Li

---

## [Author Response · Author response to Decision Letter 0]

10 Jun 2022

May 2022

Dear Editor

The revised version of the manuscript entitled "Association between the risk of malnutrition and functional capacity in patients with peripheral arterial disease: A cross-sectional study" is attached.

We have revised the manuscript following the reviewers' comments and have included changes to the manuscript highlighted in blue text in the marked version of the manuscript. We hope you find this article suitable for publication. Thank you for your time and consideration.

Sincerely,

Reviewer #1: 1. The authors are commended for their work.

2. I still continue to see the term "intermittent claudication" used in some (but not all) vascular publications. The nomenclature using the term "intermittent" seems at least to this reviewer, to be outdated. The ACC/AHA guidelines do not refer to it in this way and we don't typically refer to angina as "intermittent angina" so consider just referring to the symptom as "claudication."

Thanks for your comments. We revised and updated the term "claudication" as requested.

3. Abstract, line 40: Revise the sentence "We considered as significant p<0.05" as it is awkward as written.

Yes, we change the sentence objectively.

4. Abstract, line 42: Do the authors mean distance achieved during the six-minute walk test, claudication onset time? Please clarify absolute and relative as well, typically six-minute walk test distance is a key performance outcome often reported.

Thanks for the questions. We analyzed the six-minute walk test as the maximal distance (absolute) achieved at the end of the test. Further, we have recently expanded the possibility of analysis of the six-minute walk by adjusting the results for a healthy subject (relative) with similar characteristics (Please see: PMID: 32800882). 

5. Abstract, line 43: Lower SPPB total score?

Yes, we found significantly lower scores in the SPPB total score in patients at risk of malnutrition and when we analyzed the SPPB separated by all tests. We modified this expression in the text.

6. Risk of malnutrition and peripheral arterial disease are already in the title, so consider different keywords.

Yes, we changed the keywords as requested by the reviewer. 

7. Introduction, line 62: Minor but change "lower limbs strengths" to "lower limb strength".

Yes, we corrected the term.

8. Were other comorbidities exclusionary? For example, patients with heart failure, previous cardiac and/or peripheral revascularization, those taking medications to treat PAD (cilostazol, etc.) were excluded? These and other comorbidities could have had an impact on functional testing so please clarify or perform analyses controlling for these variables.

Yes, we reanalyzed and inserted the variables suggested by the reviewer in the manuscript table. 

9. For the six-minute walk test, why were patients asked to complete it at their usual pace, rather than as fast as they could?

Because the main limitation of PAD is the pain in the lower limbs during walking, we recommend patients walk at the usual pace to avoid them stopping earlier during the test. In addition, they also were advised by the researchers that the goal of the six-minute walk test is to achieve the greatest distance possible by walking back and forth along a 30 m corridor for six minutes, which is in accordance with the 6MWT recommendation for PAD (please see: 10.1161/CIRCULATIONAHA.114.007002)

10. Discussion, lines 225-226: Please revise "worst perfusion".

Yes, we changed the text and better explained the perfusion time for the vascular aspect. 

11. Table 1: Minor, but there is a typo (Phycological).

Thank you, we corrected the word.

12. I appreciate Table 2 and the information it provides. Still, I think it would be helpful for the readership to give a figure(s) to display some of the data, if possible, graphically.

Thank you for your suggestion. Because we used logistic regression analysis, we believe that a table is the most suitable representation method to inform the reader

13. The paper would benefit from additional copy editing for the English language.

Yes, we will insert an editable copy in the English language.

Reviewer #2: The manuscript studies the association between the risk of malnutrition and functional capacity in patients with peripheral arterial disease and claudication. The disease process and associated co-morbidities can lead to poor quality diet and low levels of functional capacity sometimes complete immobility, with resultant energy, protein, and micronutrient deficiencies. This research attempts to emphasize that a proper diet maybe improves the functional capacity of patients with peripheral arterial disease and claudication, which implies dietetic care plays a vital role in the management of PAD. Before I can recommend it for publication, the following questions and comments should be addressed.

1)An review published in 2020 year demonstrated that most patients with PAD are overweight or obese, 3/4 under sub-optimal nutritional status and high-fat mass, lower vitamins, and minerals. I suggest the authors explain the definition of malnutrition in people with PAD clearly.

Thanks for the question. When we seek to better understand malnutrition and PAD, we found that malnutrition, according to the World Health Organization, the term "malnutrition" is related to both the lack of essential nutrients and the imbalance or excess in the intake of nutrients such as carbohydrates. and saturated fats. Yes, we agree with your quote, we found in the studies with the PAD that the patients were overweight, but in the food and/or blood analysis, they had significant nutrient deficiencies that impacted their functional capacity. Low body weight is considered an indicator of malnutrition, let's say "more noticeable", but other factors must be analyzed. We did not find significant differences between the groups regarding weight or BMI.

2)I notice that the SPPB consists of a series test. I suggest the authors explain it in the abstract section briefly. Now, the methods part and the results part are not a one-to-one correspondence.

Yes, we have inserted this item in the abstract. (Line 34)

3)The authors only concluded the association between the risk of malnutrition and objective measurements of functional capacity in patients with peripheral arterial disease and claudication. How about the association between the risk of malnutrition and subjective measures of functional capacity in patients with peripheral arterial disease and claudication?

We did not find significant statistical differences in the subjective measures of functional capacity, which could be explained by the fact that the patient's main limiting factor is the pain when they are walking, thus underestimating their functional capacity in the questionnaire responses. When objectively evaluated, they presented better performance.

4)For the relationship between functional capacity and PAD, I suggest the authors give an explicit explanation.

The presence of claudication symptoms decreases the walking ability of patients with PAD. Pain during walking makes the patient seek strategies to avoid the symptoms of claudication, which is usually done by reducing physical activity levels. A sedentary lifestyle, in turn, impairs the components of physical fitness and indicators of quality of life. In addition, these limitations can lead to loss of independence, increased hospitalization and mortality rates.

5)For the relationship between nutritional status and PAD, I suggest the authors give an explicit explanation.

As demonstrated in studies on PAD, inadequate nutrition may promote the progression of the inflammatory process on the epithelium and changes to the immune system that contributes to atherosclerotic plaque development. And atherosclerotic narrowing of the lower limbs' arteries but related to reduced muscle perfusion and reduced oxygenation that can change skeletal muscle function in density, contractility, and strength that compromise mobility. We have included this explanation in the discussion section. 

6)For the applicability of MNA-SF in brazil’s PAD, I suggest the authors give an explicit explanation.

The MNA-SF in Brazil is a validated questionnaire, easy to apply and understand by the patient, and the average application time is only three minutes. We were able to use it during the patient's consultation at the Hospital.

7)In lines 72-79, it is not clear why the authors want to invest the association between the risk of malnutrition and functional capacity in patients with peripheral arterial disease and claudication. I suggest the authors give an explicit explanation.

We choose to study malnutrition because previous studies have been showing that impaired nutritional status has been considered an additional risk factor for the severity of PAD. For example, a study conducted by McDermott demonstrated that deficiency in vitamin D was associated with lower muscle density and poor functional capacity. In another study conducted by Gardner et al. the diet of patients with PAD was monitored according to their national nutritional recommendations and it was shown that about 35% followed an average of 50% of the recommendations and none followed the recommendations completely and low nutrient intake was significantly associated with shorter distances in the six-minute walk test. Thus, malnutrition may be considered a risk factor which can promote a more significant function decline in PAD patients. 

8)In lines 99-101, the inclusion criteria are inconsistent with the abstract section. How about the assessment of claudication？

Yes, we added the inclusion criteria in the abstract as the manuscript as suggested by the reviewer. We did not include any questionnaire to assess claudication symptoms. 

9)In lines 202-206, the results section, it is not clear the association between the risk of malnutrition and subjective measurements of functional capacity (WIQ and WELCH) in patients with peripheral arterial disease and claudication, and the balance with the risk of malnutrition.

We did not find an association between subjective measures (WIQ and WELCH) and balance and risk of malnutrition. We believe that patients have difficulty with self-perception in reporting their functional capacity and balance because it is a relatively easy test where the patient needs to remain in the position for 10 seconds. Most patients were able to perform this test.

10)In lines 225-228, it is not clear how nutritional deficiency aggravates the functional capacity in people with PAD and claudication.

In the figure below, we seek to explain the possible mechanisms involved in the worsening of PAD in patients at risk of malnutrition. We included the possible mechanisms in the discussion section. 

11)In lines 244-246, there are some wrong descriptions. it is not clear whether functional capacity is due to nutritional deficiency in this study.

Thank you for the question. Functional capacity is impaired in patients with PAD, and this finding is established in the literature, but we believe that nutritional status may have an additional factor in this impairment. 

12)In addition, I suggest the authors improve the presentation by considering the following minor changes:

Line 40, “per cent” -> “percent”,

Line 73, the questionnaire used in references 13 and 14 should be MNA instead of MAN-SF. 

Yes, we made the corrections.

---

## [Decision Letter · Decision Letter 1]

4 Jul 2022

PONE-D-22-05299R1Association between the risk of malnutrition and functional capacity in patients with peripheral arterial disease: A cross-sectional studyPLOS ONE

Dear Dr. Cucato,

Thank you for submitting your manuscript to PLOS ONE. After careful consideration, we feel that it has merit but does not fully meet PLOS ONE’s publication criteria as it currently stands. Therefore, we invite you to submit a revised version of the manuscript that addresses the points raised during the review process.

We look forward to receiving your revised manuscript.

Kind regards,

Yih-Kuen Jan, PhD

Academic Editor

PLOS ONE

Journal Requirements:

Reviewers' comments:

Reviewer's Responses to Questions

**Comments to the Author**

1. If the authors have adequately addressed your comments raised in a previous round of review and you feel that this manuscript is now acceptable for publication, you may indicate that here to bypass the “Comments to the Author” section, enter your conflict of interest statement in the “Confidential to Editor” section, and submit your "Accept" recommendation.

Reviewer #1: All comments have been addressed

Reviewer #2: All comments have been addressed

2. Is the manuscript technically sound, and do the data support the conclusions?

Reviewer #1: Yes

Reviewer #2: Yes

3. Has the statistical analysis been performed appropriately and rigorously? 

Reviewer #1: Yes

Reviewer #2: Yes

4. Have the authors made all data underlying the findings in their manuscript fully available?

Reviewer #1: No

Reviewer #2: Yes

5. Is the manuscript presented in an intelligible fashion and written in standard English?

Reviewer #1: No

Reviewer #2: Yes

6. Review Comments to the Author

Reviewer #1: 1. Thanks for addressing my comments.

2. Abstract line 45: Typo, change “analyzes” to “analyses.”

3. Introduction: PAD is understood to be referring to atherosclerotic plaque development in the lower limbs. Thus consider changing the opening sentence to “…which results in partial or total obstruction in the arteries of the lower limbs.” and delete “…especially in the lower extremities.”

4. Please include a hypothesis at the end of the Introduction section.

5. Page 5, line 10: “were” is repeated please delete.

6. Page 5 and 6, lines 130-131: Delete the level of accuracy of height and weight measurements. It is awkward as written so if you do want to keep it please revise to clarify.

7. Page 6 lines 141-142: Bad grammar in the sentence starting with “The data are…” Please revise.

8. Page 6 line 151: Why is there a bracket at the end of the sentence ending in “…stand test.”?

9. Page 7 line157: I thought both gender were included so please change the term “his” to include both. Also in the same sentence the statement “…and the time was timed.” is awkward. Did the authors mean and the total time was recorded?

10. “His” seems to continue to be used which needs to be changed.

11. Line 219: missing a period.

12. It would be helpful for the raw/continuous values of the outcomes were reported (e.g., 6MWT, SPPB, WIQ scores, etc).

Reviewer #2: Comments to the Authors

The authors are commended for their work.

For comments 4-6, I suggest the authors make some supplements in the introduction section.

7. PLOS authors have the option to publish the peer review history of their article (what does this mean?). If published, this will include your full peer review and any attached files.

Reviewer #1: No

Reviewer #2: No

---

## [Author Response · Author response to Decision Letter 1]

28 Jul 2022

Dear Reviewers

We have revised the manuscript following the reviewers' comments and have included changes to the manuscript and hope to consider it suitable for publication.

Reviewer #1: 

2. Abstract line 45: Typo, change “analyzes” to “analyses.”

Thank you very much for the observation. It has been changed in the text. (Line 43).

3. Introduction: PAD is understood to be referring to atherosclerotic plaque development in the lower limbs. Thus consider changing the opening sentence to “…which results in partial or total obstruction in the arteries of the lower limbs.” and delete “…especially in the lower extremities.”

We have changed it according to the reviewer's suggestion. (Line 57).

4. Please include a hypothesis at the end of the Introduction section.

As suggested, we included the study hypothesis at the end of the introduction. (Line 81).

5. Page 5, line 10: “were” is repeated please delete.

Yes, we deleted it. (Line 105 - 107).

6. Page 5 and 6, lines 130-131: Delete the level of accuracy of height and weight measurements. It is awkward as written so if you do want to keep it please revise to clarify.

We followed the reviewer's important suggestion and removed it from the text.

7. Page 6 lines 141-142: Bad grammar in the sentence starting with “The data are…” Please revise.

Thank you, we modified the text. (Line 133 – 136).

8. Page 6 line 151: Why is there a bracket at the end of the sentence ending in “…stand test.”?

Yes, we changed. (Line 143).

9. Page 7 line157: I thought both gender were included so please change the term “his” to include both. Also in the same sentence the statement “…and the time was timed.” is awkward. Did the authors mean and the total time was recorded?

Thank you for this consideration. We have revised the grammar of this paragraph. (Line 147 – 149).

10. “His” seems to continue to be used which needs to be changed.

Yes, we changed. 

11. Line 219: missing a period.

Thank you, we changed. (Line 211).

12. It would be helpful for the raw/continuous values of the outcomes were reported (e.g., 6MWT, SPPB, WIQ scores, etc).

Thanks for the comment. The 6MWT is presented in raw and relative data. However, the WIQ and SPPB tests are scored based on calculations.

Reviewer #2: Comments to the Authors

The authors are commended for their work.

For comments 4-6, I suggest the authors make some supplements in the introduction section.

We appreciated the reviewer's suggestions and made the necessary changes.

---

## [Decision Letter · Decision Letter 2]

2 Aug 2022

Association between the risk of malnutrition and functional capacity in patients with peripheral arterial disease: A cross-sectional study

PONE-D-22-05299R2

Dear Dr. Cucato,

We’re pleased to inform you that your manuscript has been judged scientifically suitable for publication and will be formally accepted for publication once it meets all outstanding technical requirements.

Kind regards,

Yih-Kuen Jan, PhD

Academic Editor

PLOS ONE

Additional Editor Comments (optional):

Reviewers' comments:

Reviewer's Responses to Questions

**Comments to the Author**

1. If the authors have adequately addressed your comments raised in a previous round of review and you feel that this manuscript is now acceptable for publication, you may indicate that here to bypass the “Comments to the Author” section, enter your conflict of interest statement in the “Confidential to Editor” section, and submit your "Accept" recommendation.

Reviewer #1: All comments have been addressed

Reviewer #2: All comments have been addressed

2. Is the manuscript technically sound, and do the data support the conclusions?

Reviewer #1: Yes

Reviewer #2: Yes

3. Has the statistical analysis been performed appropriately and rigorously? 

Reviewer #1: Yes

Reviewer #2: Yes

4. Have the authors made all data underlying the findings in their manuscript fully available?

Reviewer #1: Yes

Reviewer #2: Yes

5. Is the manuscript presented in an intelligible fashion and written in standard English?

Reviewer #1: Yes

Reviewer #2: Yes

6. Review Comments to the Author

Reviewer #1: (No Response)

Reviewer #2: (No Response)

7. PLOS authors have the option to publish the peer review history of their article (what does this mean?). If published, this will include your full peer review and any attached files.

Reviewer #1: No

Reviewer #2: No

---

## [Editor Report · Acceptance letter]

22 Aug 2022

PONE-D-22-05299R2 

Association between the risk of malnutrition and functional capacity in patients with peripheral arterial disease: A cross-sectional study 

Dear Dr. Cucato:

I'm pleased to inform you that your manuscript has been deemed suitable for publication in PLOS ONE. Congratulations! Your manuscript is now with our production department. 

Kind regards, 

on behalf of

Dr. Yih-Kuen Jan 

Academic Editor

PLOS ONE